# Role of PLEXIND1/TGFβ Signaling Axis in Pancreatic Ductal Adenocarcinoma Progression Correlates with the Mutational Status of *KRAS*

**DOI:** 10.3390/cancers13164048

**Published:** 2021-08-11

**Authors:** Sneha Vivekanandhan, Vijay S. Madamsetty, Ramcharan Singh Angom, Shamit Kumar Dutta, Enfeng Wang, Thomas Caulfield, Alexandre A. Pletnev, Rosanna Upstill-Goddard, Yan W. Asmann, David Chang, Mark R. Spaller, Debabrata Mukhopadhyay

**Affiliations:** 1Department of Biochemistry and Molecular Biology, Mayo Clinic College of Medicine and Science, Jacksonville, FL 32224, USA; vivekanandhan.sneha@mayo.edu (S.V.); madamsetty.vijay@mayo.edu (V.S.M.); angom.ramcharan@mayo.edu (R.S.A.); Dutta.Shamit@mayo.edu (S.K.D.); Wang.Enfeng@mayo.edu (E.W.); caulfield.thomas@mayo.edu (T.C.); 2Department of Chemistry, Dartmouth College, Hanover, NH 03755, USA; Alexandre.pletnev@dartmouth.edu (A.A.P.); Mark.R.Spaller@Dartmouth.edu (M.R.S.); 3Wolfson Wohl Cancer Research Centre, Institute of Cancer Sciences, University of Glasgow, Garscube Estate Switchback Road, Glasgow G12 8QQ, UK; Rosie.Upstill-Goddard@glasgow.ac.uk (R.U.-G.); David.Chang@glasgow.ac.uk (D.C.); 4Health Sciences Research, Mayo Clinic College of Medicine and Science, Jacksonville, FL 32224, USA; Asmann.Yan@mayo.edu; 5Geisel School of Medicine at Dartmouth and Norris Cotton Cancer Center, Lebanon, NH 03756, USA; 6Division of Natural and Applied Sciences, Duke Kunshan University, Kunshan 215316, China

**Keywords:** pancreatic ductal adenocarcinoma, PLEXIND1, KRAS, TGFβ, SMAD

## Abstract

**Simple Summary:**

Pancreatic cancer is among the most lethal cancers. The expression of PLEXIND1, a receptor, is upregulated in many cancers (including pancreatic cancer). Traditionally, PLEXIND1 is known to be involved in neuron development and mediate semaphorin signaling. However, its role and signaling in cancer is not fully understood. In our study, we present a new mechanism through which PLEXIND1 mediates its roles in cancer. For the first time, we demonstrate that it can function as a transforming growth factor beta coreceptor and modulate SMAD3 signaling. Around 90% of pancreatic cancer patients have mutant KRAS. Our work suggests that PLEXIND1 functions differently in pancreatic cancer cell lines, and the difference correlates with KRAS mutational status. Additionally, we demonstrate a novel peptide based therapeutic approach to target PLEXIND1 in cancer cells. Our work is valuable to both neuroscience and cancer fields, as it demonstrates an association between two previously unrelated signaling pathways.

**Abstract:**

PLEXIND1 is upregulated in several cancers, including pancreatic ductal adenocarcinoma (PDAC). It is an established mediator of semaphorin signaling, and neuropilins are its known coreceptors. Herein, we report data to support the proposal that PLEXIND1 acts as a transforming growth factor beta (TGFβ) coreceptor, modulating cell growth through SMAD3 signaling. Our findings demonstrate that PLEXIND1 plays a pro-tumorigenic role in PDAC cells with oncogenic KRAS (KRAS^mut^). We show in KRAS^mut^ PDAC cell lines (PANC-1, AsPC-1,4535) PLEXIND1 downregulation results in decreased cell viability (in vitro) and reduced tumor growth (in vivo). Conversely, PLEXIND1 acts as a tumor suppressor in the PDAC cell line (BxPC-3) with wild-type KRAS (KRAS^wt^), as its reduced expression results in higher cell viability (in-vitro) and tumor growth (in vivo). Additionally, we demonstrate that PLEXIND1-mediated interactions can be selectively disrupted using a peptide based on its C-terminal sequence (a PDZ domain-binding motif), an outcome that may possess significant therapeutic implications. To our knowledge, this is the first report showing that (1) PLEXIND1 acts as a TGFβ coreceptor and mediates SMAD3 signaling, and (2) differential roles of PLEXIND1 in PDAC cell lines correlate with KRAS^mut^ and KRAS^wt^ status.

## 1. Introduction

Pancreatic ductal adenocarcinoma (PDAC) is among the leading causes of cancer-related deaths [1]. The five-year survival rate is only 5–7%, and fewer than 20% of patients achieve one-year survival. One of the contributing factors for this is the heterogeneity in the tumor that facilitates the deregulation of several signaling pathways and renders therapies targeting cancer ineffective [2]. It is therefore crucial that new signaling pathways be identified and characterized so that new molecular therapeutic targets can be discovered.

One such potential target is PLEXIND1. Typically, the expression of PLEXIND1 is low in adult tissues [3] and is thought to be limited to a subset of activated fibroblasts and macrophages [4]. However, in several types of cancers (including pancreatic cancer), PLEXIND1 is overexpressed in both tumor cells and their vasculature [5]. Consequently, PLEXIND1 is gaining prominence in cancer research.

PLEXIND1 can act as both a tumor promoter and tumor suppressor [6]. Some of the potential reasons for these opposing roles of PLEXIND1 in cancer are paracrine versus autocrine signaling [7]; differences in signaling response stimulated by various ligands and their different isoforms [8]; variances in the cell and tissue type [9]; and association with different proteins that can alter the signaling outcome [10].

TGFβ signaling is often deregulated in PDAC, and it can both suppress and promote tumor growth [11]. In prior publications, we and others have demonstrated that Neuropilin 1 (NRP1) can function as a TGFβ coreceptor [12,13]. As NRP1 is an established coreceptor for PLEXINs [14,15], this prompted us to examine whether PLEXIND1 could also function as a TGFβ coreceptor. Herein, we report a novel finding that in KRAS^mut^ PDAC cell lines PLEXIND1 is pro-tumorigenic and mediates SMAD3 signaling. Conversely, in a KRAS^wt^ PDAC cell line, BxPC-3, PLEXIND1 is anti-tumorigenic. To our knowledge, this is the first study that has shown that the role of PLEXIND1 in cancer correlates with the mutational status of *KRAS*. These findings elucidate a previously unknown PLEXIND1-mediated signaling pathway and a novel cause for the dual role PLEXIND1 in at least some forms of cancer growth.

## 2. Results

### 2.1. Expression Pattern and Functional Role of PLEXIND1 in PDAC

In evaluating the clinical relevance of PLEXIND1 expression in PDAC patients, we found that the higher expression of PLEXIND1 correlated with lower survival probability compared to tumors with lower expression levels (Figure 1A). The median expression value of PLEXIND1 was used to stratify samples (n-94) into high and low expression groups, with the 50% of samples with PLEXIND1 expression values above the median assigned to the high group and the 50% of samples with PLEXIND1 expression values below the median assigned to the low group. Next, we examined the expression of PLEXIND1 in three commercial and five patient-derived PDAC cell lines. Seven of these eight cell lines expressed PLEXIND1, albeit at different levels (Figure 1B). Our findings corroborate a published report where the majority of the PDAC cells evaluated expressed PLEXIND1 [5].

To study the effect of reduced PLEXIND1 expression in PDAC cell lines, we employed shRNA-mediated knockdown in PANC-1 and AsPC-1 cell lines (Figure 1C,E). On assessing the cell viability 72 h post-PLEXIND1 knockdown, we observed that it was notably lower in cells with reduced PLEXIND1 levels compared to their controls (Figure 1D,F). These results suggest that PLEXIND1 plays a role in maintaining cell viability in in vitro models for PANC-1 and AsPC-1 cell lines. Next, we used a Tet-Inducible CRISPR/CAS9 (sgRNA)-mediated PLEXIND1 knockdown system in PANC-1 cells and obtained similar results (Appendix A). However, an interesting finding in the sgRNA- system was that many of the housekeeping proteins (including GAPDH, beta-actin, alpha-tubulin, vinculin, and 18S rRNA) were downregulated (data not shown; two different sgRNAs were tested).

To understand how cell viability was reduced, we examined the expression of p21, a mediator of cell cycle arrest [16]. We found that it was downregulated in cells with reduced PLEXIND1expression (Appendix A). Some studies have reported that p21 can function as an oncogene [17,18], and this seems to be a possibility in our model. We then studied the expression of E-cadherin (E-cad), a tumor suppressor protein involved in cell–cell adhesion [19], and found its expression to be upregulated in cells with PLEXIND1 knockdown relative to control cells (Appendix A). This is in line with published literature suggesting that the loss of E-cad in PDAC patients is associated with a worsened median survival, compared to patients with normal E-cad expression [20]. Interestingly, it has been demonstrated in a prostate cancer model that PLEXIND1 enhances cell migration and downregulates E-cad expression in a slug-mediated manner [3]. These data support our contention that PLEXIND1 is pro-tumorigenic in PANC-1 cells and its expression influences E-cad expression.

### 2.2. Role of PLEXIND1 in PDAC Progression

We next investigated the effect of reduced PLEXIND1 expression using orthotopic mice models. Tumors derived from mice implanted with cells treated with PLEXIND1 shRNA (Figure 2A; Appendix A) had lower volumes (Figure 2B; Appendix A) and weight (Figure 2C) than their respective controls. We observed the same trend in tumors obtained from mice injected with cells treated with control or PLEXIND1 sgRNA (Appendix A). Similarly, the number of cells that stained positive for Ki-67 were lower in tissues derived from tumors harvested from mice implanted with cells with reduced PLEXIND1 expression, relative to their control counterparts (Figure 2D). These findings corroborate our in vitro results that PLEXIND1 is involved in for tumor growth.

NRP1, a PLEXIND1 coreceptor [21], can function as a TGFβ coreceptor [12,22]. Furthermore, while the canonical ligand of PLEXIND1 is Semaphorin 3E (Sema 3E) [23], a study reported that in human melanoma, PLEXIND1 aided the invasive and metastatic nature of cancer and Sema 3E was not the activating ligand [24]. A recent study reported that microRNA-27b positively promoted TGFβ-mediated endothelial–mesenchymal transition and regulated PlexinD1 expression in mouse pancreatic microvascular endothelial cells [25]. These led us to investigate whether PLEXIND1 could function as a TGFβ coreceptor.

To explore this possibility, we performed co-immunoprecipitation assays with TGFβ induction. We observed that PLEXIND1 and TGFβRII were present in the same immunocomplex in PANC-1 cell lysates. Additionally, PLEXIND1 protein expression levels increased after 10 and 30 min of TGFβ stimulation (Figure 3A–C). To our knowledge, this is the first reported study in which PLEXIND1 is present in the same immunocomplex as TGFβRII, and that TGFβ influences PLEXIND1 protein expression.

We then conducted modeling analysis to determine whether PLEXIND1 could bind TGFβ and TGFβRII in the absence of NRP1. The model is shown (Figure 3D) using the PIPER program within the Schrödinger software suite. These results show that the proteins PLEXIND1, TGFβRII, and TGFβ associate at a common region on PLEXIND1, which acts as the host for the proteins to form a coordinated complex that is stabilized as a ternary complex.

Next, we wanted to probe whether PLEXIND1 expression influenced NRP1 expression, and found that, in PANC-1 cells, NRP1 protein expression did not change immediately (within a week). However, when the cells were maintained in culture for an extended time period (over a few weeks, with partial PLEXIND1 knockdown), the protein expression of NRP1 was upregulated (Appendix A).

Our data thus far have demonstrated that the absence of PLEXIND1 inhibits tumor growth and PLEXIND1 downregulation in PANC-1 cells causes an increase in NRP1 protein expression. We previously reported that NRP1 acted as a tumor suppressor in PANC-1 cells [26], in accordance with our current findings.

### 2.3. PLEXIND1 Modulates SMAD3 Signaling and, Eventually, PDAC Growth

To gain a better understanding of how PLEXIND1 could function as a TGFβ coreceptor, we studied the involvement of SMADs in our model. PANC-1, AsPC-1, and 4535 (a patient-derived PDAC cell line) cells with shRNA-mediated (Figure 4A–C) and sgRNA-mediated (Appendix A) PLEXIND1 knockdown were found to reduce protein levels of phosphorylated and total SMAD3. Some PDAC cells are difficult to transfect, and in AsPC-1 cells, the PLEXIND1 shRNA1 yielded minimal knockdown. Additionally, TGFβ stimulation partially restored the expression of total SMAD3 in PANC-1 cells with PLEXIND1 knockdown (Figure 4A). A report demonstrated that SMAD3 facilitates tumor growth in pancreatic cancer [27]. Our finding thus appears to support the paradigm that SMAD3 is pro-tumorigenic.

PANC-1 and AsPC-1 cells with SMAD3 knockdown were then generated. Notably, in these cells, the protein levels of PLEXIND1 were reduced as well (Figure 4D,E). We observed no change in PLEXIND1 protein expression in 4535 cells with SMAD3 knockdown (Appendix A). To further examine the role of SMAD3 in our tumor model, we implanted PANC-1 cells with SMAD3 knockdown in orthotropic mice models. The tumors derived from mice that were injected with PANC-1 cells with SMAD3 knockdown had lower volumes and weights compared with those derived from corresponding control mice (Figure 4F,G). Furthermore, immunohistochemical analysis on tissues showed that tumors derived from PANC-1 cells with SMAD3 knockdown had fewer Ki-67 positive cells as compared to their controls, suggesting reduced proliferation (Appendix A). These findings indicate that SMAD3 likely plays a role in PLEXIND1-mediated tumor development, and fits with our model; namely, in PANC-1 cells, PLEXIND1 knockdown leads to SMAD3 downregulation, and the lack of both PLEXIND1 and SMAD3 individually impair tumor growth and development.

We have previously published that PANC-1 cells with SMAD2 knockdown have increased cell viability and tumor growth [26]. To this we now add that in these cells, PLEXIND1 expression is increased (Figure 4H). This supports our current hypothesis that PLEXIND1 is pro-tumorigenic in PANC-1 cells. Equally, when PANC-1 cells with partial PLEXIND1 knockdown were maintained in culture for over a few weeks, the protein expression of both phosphorylated and total SMAD2 were upregulated (Appendix A).

Taken together, our data point to a connection between the expression of PLEXIND1 and SMADs2/3 and the involvement of SMAD3 in PLEXIND1-mediated tumor growth in the PANC-1 cell line. An important next step is to validate this in additional cancer types.

### 2.4. Decreased PLEXIND1 Expression Reduces RAC-1 Expression in PDAC Cells

We next looked at TGFβ/SMAD3 targets and observed a decrease in the mRNA levels of SMAD7 and Serpine [28,29] in PANC-1 cells with reduced PLEXIND1 that had been in culture for about two weeks (Figure 5A,B). Han et al. [30] reported that, in mouse keratinocytes, Smad7 increased the expression and activity of Rac-1, a Rho GTPase [31] that is upregulated in several cancers, including PDAC [32,33].

Examining the effect of PLEXIND1 downregulation on RAC-1 levels, we observed that in PANC-1 and 4535 cells with PLEXIND1 knockdown, there was a slight decrease in the RAC-1 protein expression relative to control cells (Figure 5C,D; Appendix A). Of note, we also saw a reduction in RAC-1 expression in PANC-1 and 4535 cells with SMAD3 knockdown (Appendix A). This data suggest that RAC-1 may be involved in the PLEXIND1/SMAD3-mediated tumor growth in PANC-1 and 4535 cells.

Several studies in various models have reported that RAC proteins facilitate KRAS^mut^ mediated oncogenic transformation [33,34,35]. Over 90% of pancreatic cancers harbor KRAS^mut^ [26]. Our interest piqued, we looked at KRAS expression and found it decreased in PANC-1 and 4535 cells upon PLEXIND1 knockdown (Figure 5E,F). Similarly, there was reduced KRAS expression in PANC-1 and 4535 cells with SMAD3 knockdown (Appendix A). Another important observation was that the levels of RAC-1 and KRAS were also partially restored in PANC-1 cells with PLEXIND1 knockdown upon TGFβ stimulation (data not shown).

### 2.5. PLEXIND1 Acts as a Tumor Suppressor in KRAS^wt^ PDAC Cell Line BxPC-3

We have previously published that the genetic status of *KRAS* modulates the role of NRP1 in tumorigenesis [26]. This finding made us curious to learn whether the genetic status of *KRAS* would influence PLEXIND1-mediated tumor development. Indeed, we found that in PDAC patients, higher PLEXIND1 expression correlated with lower survival probability in tumors with KRAS^mut^ but with slightly better survival probability in tumors with KRAS^wt^ (Figure 6A). KRAS^mut^ samples were defined based on the presence of a non-silent KRAS SNV or loss of function structural variant. Samples without a KRAS variant were defined as wildtype. As with the PDAC survival analysis, the median expression values of PLEXIND1 in the KRAS^mut^ cohort (n-83) and KRAS^wt^ cohort (n-11) were used to stratify samples into high and low expression groups.

The PDAC cell line BxPC-3 with KRAS^wt^ was chosen for this part of the study. We generated BxPC-3 cells with PLEXIND1 knockdown (Figure 6B) and found that in contrast to its effect in KRAS^mut^ cells, reduced PLEXIND1 expression enhanced the cell viability of BxPC-3 cells (Figure 6C). Furthermore, these cells had a different phenotype as compared to their control counterparts. The control cells were small, round, and grew in clusters, while post-PLEXIND1 knockdown cells became elongated, spindle-shaped, and grew relatively more dispersed (Figure 6D). A recent study described that PLEXIND1 regulated the morphologic changes in newborn neurons [36]. However, we are not aware of such findings for cancer cells.

Another significant finding was that the level β-actin (a housekeeping gene) was down-regulated in BxPC-3 cells with PLEXIND1 knockdown (data not shown). Using GAPDH as our loading control, we measured the expression levels of p21 and E-cad and found that, in contrast to PANC-1 cells, the level of p21 was increased in BxPC-3 cells with PLEXIND1 knockdown, while that of E-cad was decreased (Appendix A). These data suggest that p21 might be adopting a pro-tumorigenic role in our model.

We proceeded to inject BxPC-3 cells with decreased PLEXIND1 into mice (female, 6–8 weeks, SCID). Our in vivo results corroborated our in vitro data. We found that tumors from mice implanted with BxPC-3 cells with PLEXIND1 knockdown had larger tumor volumes and weights compared to their respective controls (Figure 6E,F). Additionally, immunohistochemical analysis on tissues revealed that tumors derived from BxPC-3 cells with lower PLEXIND1 expression had twice the number of Ki-67 positive cells as compared to their controls, supporting a higher proliferation rate (Appendix A).

Next, we examined and found an increase in the levels of phosphorylated SMAD3 in the tissues from mice injected with BxPC-3 PLEXIND1 knockdown cells compared to their controls (Figure 6G). Additionally, we observed an increase in the number of ducts in these tissues compared to the tumor tissues from the control mice, and the cells positive for phosphorylated SMAD3 were mainly clustered around these ducts. Nevertheless, one discrepancy is that we did not observe a change in the protein expression of phosphorylated and total SMAD3 in BxPC-3 cells with PLEXIND1 knockdown compared to their controls (data not shown). A possible explanation is that the cells were not maintained in culture for as long as they were in the in vivo experiments.

Following this, we studied the expression of phosphorylated and total SMAD2 in BxPC-3 cells with reduced PLEXIND1 and found no change compared to controls (Appendix A). We next studied the PLEXIND1 protein expression levels upon TGFβ stimulation. In contrast to our findings in KRAS^mut^ PANC-1 cells, here the expression was downregulated after 10 and 30 min of TGFβ stimulation (Appendix A).

The expression of NRP1 was then measured and found to be upregulated in BxPC-3 cells with PLEXIND1 knockdown (Appendix A). This supports our earlier work, where we demonstrated that NRP1 downregulation in BxPC-3 cells leads to decreased cell viability and tumor growth [26]. After NRP1, we assessed the protein levels of RAC-1 and KRAS^wt^; to our surprise, we found that they were downregulated in BxPC-3 cells with PLEXIND1 knockdown (Appendix A). One tentative explanation for this downregulation of KRAS^wt^ is that it could be acting as a tumor suppressor. There have been reports in which wild-type KRAS proteins have been demonstrated to be anti-tumorigenic [37,38]. These results suggest that PLEXIND1 may be utilizing a different downstream mechanism to facilitate tumor growth in BxPC-3 cells.

We then considered the proliferation marker ERKp44/42. A study in endometrioid ovarian cancer reported that ERK/MAPK and PI3K activation enhanced Sema 3E/PLEXIND1-induced EMT through nuclear localization of Snail1 [39]. We observed that, while the phosphorylated form was upregulated (in line with tumor growth), the total form was unexpectedly downregulated (Appendix A).

Collectively, this set of data indicates that PLEXIND1 acts as a tumor suppressor in BxPC-3 cells. In the absence of PLEXIND1, there is a morphological change in the BxPC-3 cells, and these cells have a more aggressive phenotype compared to their control cells. Furthermore, while PLEXIND1 potentially engages SMAD3 and NRP1 to mediate tumor growth in BxPC-3 cells, the involvement of RAC-1 and KRAS is unclear.

### 2.6. Therapeutic Potential of Peptide-Mediated Targeting of PLEXIND1

Selected reports within the published literature suggest that PLEXIND1 or the signaling pathways it mediates could serve as therapeutic targets [3,4,7]. Our results also suggest that therapeutically targeting PLEXIND1-mediated interactions, specifically in KRAS^mu*t*^ cancer cells, could impair tumor growth and development. As a starting point towards a targeted approach, we sought to develop a molecular agent that could specifically disrupt or inhibit PLEXIND1-mediated interactions.

We used computational modeling studies to understand how peptide sequences would potentially bind PLEXIND1 and gauge their binding efficiency. Two sequences, the *N*-myr-YECYSEA (native sequence; Supp Appendix A–A’’’) and *N*-myr-AEYCESY (scrambled sequence; Supp Appendix A–B’’’) were docked with the PLEXIND1 structure. Multiple dockings were completed using a winner-take-all approach (survival) to achieve the fit docking poses from a pool of multiple sites and docking poses. The top docking pose for *N*-myr-YECYSEA was −10.17 kcal/mol*Å^2^ and *N*-myr-AEYCESY was −6.74 kcal/mol*Å^2^ (Figure 7A,B). Although we recognize that docking energetics often do not accurately correlate to those obtained from empirical binding assays, these simulations suggest the YECYSEA sequence to be about 10^3^ times better PLEXIND1 binder, given the logarithmic nature of the docking values. All peptides were amidated with myristic acid at their amino termini, a modification intended to enhance cell permeability and serum stability, and this seemed to produce some steric hindrance for adjacent molecules. This may well affect TGFβ binding despite not having any direct interactions with PLEXIND1.

To evaluate the biologic relevance of these peptides, we performed in vitro toxicity assays in PANC-1 cells. Cells treated with the PLEXIND1 C-terminal mimetic peptide, *N*-myr-YECYSEA, had an IC_50_ value of 42.2 µM, significantly lower than that for the control (scrambled) peptide (IC_50_ > 400 µM) (Figure 7C,D). Next, we developed PANC-1 sub-cutaneous tumors in SCID mice. After the tumor volumes reached 250–300 mm^3^, the mice were randomly divided into two groups. The control group was treated with vehicle (75% DMSO in water), while the treatment group received 500 µg/mice of PLEXIND1 peptide (AP1134 dissolved in 75% DMSO in water) 5d/wk for three weeks. The tumor volumes were monitored every week. From the start, tumor volumes in mice in the treatment group were lower than the tumor volumes measured in the control group mice (Figure 7E). After three weeks, the mice were sacrificed. The tumors were checked to confirm PLEXIND1 knockdown (Figure 7F). As anticipated, the volumes and weights of the tumors obtained from mice in the treatment group were significantly lower than those of the corresponding controls (Figure 7G,H).

## 3. Discussion

The role of PLEXIND1 in cancer is an emerging field, with published literature supporting its role as both a tumor promoter and a tumor suppressor. The pro-tumor role of PLEXIND1 has been reported to be mediated by p61-Sema 3E/PLEXIND1 in colon cancer [7] and Sema 3E/PLEXIND1 signaling in ovarian endometrioid cancer [39]; in prostate cancer PLEXIND1 has been shown to act as a transcriptional target stimulated by the Notch signaling that helped in Slug- mediated E-cadherin downregulation [3]. PLEXIND1, on the other hand, has been reported to promote apoptosis in association with orphan nuclear receptor NR4A1 in the absence of Sema 3E in a breast cancer mouse model [40].

Mouse pancreatic microvascular endothelial cell microRNA-27b was shown to act as a positive mediator of TGFβ facilitated endothelial –mesenchymal transition, and regulated plexinD1 expression [25]. In our study we examined the role of PLEXIND1 in a pancreatic cancer model. Our combined data support a new mechanism involving PLEXIND1/TGFβ signaling outlined in our working model (Appendix A). PLEXIND1 plays a pro-tumorigenic role in KRAS^mut^ cells, while it acts as a tumor suppressor in KRAS^wt^ cells, suggesting a correlation between the genetic status of KRAS and role of PLEXIND1 in the PDAC cells.

We have demonstrated a connection between PLEXIND1 and SMADs. In PANC-1 cells, PLEXIND1 downregulation results in a decrease in protein expression of the SMAD3 and mRNA levels of SMAD7 and Serpine1, potentially abetting tumor suppression. Furthermore, the protein levels of SMAD2 and SMAD3 impact PLEXIND1 protein levels in PANC-1 cells. There is conceivably a feedback loop that connects all these proteins, their expressions, and potentially their activities. Another possibility is that the expression of one or more of these proteins might be linked to the expression of one or more proteins that might be key regulators of (or contribute to) protein stability.

Nevertheless, while all of these findings support a close association between PLEXIND1 and TGFβ signaling, further studies are required to completely understand the links and decipher the mechanism(s) involved. It is intriguing as to how the downregulation of PLEXIND1 results in the reduced expression of many other proteins, including those with housekeeping functions. These results suggest that PLEXIND1 is likely crucial for cell viability. Additionally, we find it fascinating as to how the expression of some proteins changes within seventy-two hours, while for others it requires over two weeks. We think it is imperative to extend these findings to other cell lines and cancer types.

Our data show that PLEXIND1 knockdown downregulates tumor growth and that there is upregulation of NRP1 in KRAS^mu*t*^ cells (Figure 1 and Figure 2 and Appendix A). Furthermore, we observed an increase in the PLEXIND1 protein expression upon TGFβ stimulation in PANC-1 cells. Significantly, we previously published that NRP1 acts as a tumor suppressor in PANC-1 cells, and TGFβ induction downregulates its expression [26]. Conversely, in BxPC-3 cells, PLEXIND1 reduction enhances tumor growth, and there is upregulation of NRP1; we have earlier published that NRP1 facilitates tumor growth in BxPC-3 cells [26].

All this may indicate a new interface for PLEXIND1 and NRP1: TGFβ signaling. If correct, this novel finding of an association between PLEXIND1 and TGFβRII leads to two central questions: What cellular processes besides axonal guidance is PLEXIND1 involved in, and can it bind other TGFβ receptors? Further in-depth studies will be required to validate that PLEXIND1 is a TGFβ coreceptor.

PLEXIND1 has a serine–glutamate–alanine (SEA) sequence (part of a PDZ domain-binding motif [23]) at its carboxyl terminus that promotes binding to selected PDZ domains such as GIPC [21,41]. We previously developed a strategy of preparing lipidated peptides based on the C-terminal sequences of protein binding partners of GIPC [42]. One of the identified sequences shown to bind to GIPC was that of PLEXIND1 [23], and amongst the peptides ligands we prepared was an *N*-myristoylated peptide based on the C-terminal sequence of PLEXIND1. The use of designed peptides demonstrated a notable level of activity in a pancreatic tumor model in mice. These results provide preliminary evidence for a novel and effective targeted lipopeptide approach towards the development of new therapeutic agents in the treatment of certain pancreatic cancers.

## 4. Conclusions

Our study, to our knowledge, for the first time shows a novel finding that PLEXIND1 could act as a TGFβ receptor and modulate SMAD3 signaling in PDAC cells. Our results demonstrate that in KRAS^mut^ PDAC cells, PLEXIND1 promotes tumor growth via SMAD3 signaling. Conversely, in KRAS^wt^ PDAC cell line BxPC-3, it acts as a tumor suppressor. These data indicate a connection between the role of PLEXIND1 in PDAC cells and the mutational status of KRAS. It would be interesting to investigate this link further to completely understand this circuit and important to extend these findings to additional cancer models. Furthermore, we present an innovative strategy of disrupting PLEXIND1-mediated interactions via lipidated peptides, A few publications have disclosed that lack of PLEXIND1 results in severe defects, including lethality, in mice models [43,44]. Thus, it is essential that signaling pathways utilized by PLEXIND1 to mediate its effects in cancer be fully elucidated, as this could reveal new potential therapeutic targets. We anticipate our study will have a broad impact on cancer biology, as well as within neuroscience.

## 5. Materials and Methods

### 5.1. Survival Curves

Bulk tumor PDAC samples from the Australian Pancreatic Cancer Genome Initiative (APGI; part of International Cancer Genome Consortium) were used for all survival analyses. All data with respect to patients have been described previously and are available in an earlier publication [45]. Survival analyses were performed using the R and Bioconductor packages ‘survminer’ and ‘survcomp’ [46] with disease-specific survival as the primary endpoint.

### 5.2. Cell Culture

All commercial cell lines were purchased from ATCC. The human PC cell line PANC-1 was cultured in Dulbecco’s Modified Eagle Medium; BxPC-3 and AsPC-1 were cultured in and RPMI 1640 and the patient-derived cell line 4666 was maintained in DMEM F12 media, respectively, with each media containing 10% FBS, 1% antibiotic–antimycotic (anti–anti; Gibco, Carlsbad, CA USA), and 0.02% plasmocin (Invivogen, San Diego, CA USA). Cells were serum starved in raw media for 18 h before stimulation with TGFβ (Biolegend, San Diego, CA, USA).

### 5.3. shRNA and Tet Inducible CRISPR/CAS9 System (sgRNA) Mediated Transfections

All of the RNA and CRISPR system reagents were purchased from Dharmacon (Lafayette, CO, USA).

### 5.4. shRNA Transfection

Lentiviruses for PLEXIND1 shRNA and control shRNA were prepared as described earlier [47]. The protocol followed for transfection in target cells: Day 1, plate the cells; Day 2, add the virus along with polybrene; Day 3, change the media; Day 4, serum starvation and addition 2 μg/mL of puromycin to the medium for selecting transfected cells; and Day 5, lysate collection. In PANC-1 cells, reduced PLEXIND1 expression decreased cell viability, and the cells were not passaged or kept in culture. In contrast, PLEXIND1 knockdown enhanced the cell growth in BxPC-3 cell line and the cells were cultured. Control cell lines for each were cultured exactly like the corresponding PLEXIND1 shRNA-transfected cell lines. The plasmids for the PLEXIND1 and SMAD3 shRNA were purchased from the Mayo Clinic Core Facility (Jacksonville, FL, USA) and the control shRNA plasmid was bought from Open Biosystems (Lafayette, CO, USA).

The sequences of the shRNA used were:

PLEXIND1 shRNA1: 5′-TGCTGTTGACAGTGAGCGACCCATGACAGTCATGGTCTATTAGTGAAGCCACAGATGTAATAGACCATGACTGTCATGGGCTGCCTACTGCCTCGGA-3′

PLEXIND1 shRNA2: 5′-TGCTGTTGACAGTGAGCGAGCCAGTGGACTTCTTCATCAATAGTGAAGCCACAGATGTATTGATAAGAAGTCCACTGGCGTGCCTACTGCCTCGGA-3′

SMAD3 shRNA1:

5′-CCGGCATCTCCTACTACGAGCTGAACTCGAGTTCAGCTCGTAGTAGGAGATGTTTTT-3′

SMAD3 shRNA2:

5′-CCGGGAGCCTGGTCAAGAAACTCAACTCGAGTTGAGTTTCTTGACCAGGCTCTTTTT-3′

### 5.5. Control shRNA: 5′-GGATAATGGTATTGAGATGG-3′ Tet Inducible CRISPR/CAS9 (sgRNA) System

Edit-R Lentiviral Cas9 Nuclease Expression vectors containing a human codon-optimized version of the *S. pyogenes* Cas9 (csn1) gene and sgRNAs that were provided as plasmid DNA with blasticidin resistance marker were purchased from Horizon Inspired Cell Solutions, Dharmacon, USA (Lafayette, CO).

We synthesized the plasmid for both sgRNA and Cas9 as described by the manufacturer’s protocol, followed by lentiviral packaging using the standard protocol in the provider’s manual. Briefly, 4 µg of the target plasmid, 1ug of VsVg, and 3ug of Pax2 packaging vectors were used for viral packaging using 293T cells.

The PANC-1 and AsPC-1 cells were infected with the viral particles using the transfection protocols provided by the manufacturers. The cells were first infected with Cas9 expressing viral particles for 72 h followed by blasticidin clonal selection. The selected clones were seeded in 60 mm dishes. Cells at 60–70% confluence were infected with PLEXIND1 sgRNA expressing viral particle for 72 h and selected for puromycin resistance (1µg/mL). The puromycin-resistant cells were similarly propagated and used for the gene editing experiments. The PLEXIND1 knockout was achieved by doxycycline treatment (1µg/mL).

### 5.6. Cell Proliferation Assays

We seeded 3000 cells per well in the 96-well plates, which were grown for 72 h in complete medium. Post 72 h, the MTS cell proliferation assay (Cell Titer 96 Aqueous One Solution Cell Proliferation Assay [MTS] Promega, Madison, WI, USA) was performed.

### 5.7. Antibodies

Western blot antibodies for Ki-67, TGFBRII, KRAS, and horseradish peroxidase-conjugated secondary antibodies were purchased from Santa Cruz Biotechnology (Dallas, TX, USA); antibody against PLEXIND1 from R&D Biosciences (Minneapolis, MN, USA) and Abcam (Cambridge, MA, USA); antibodies against GAPDH and total Smad3 from Cell Signaling Technology Inc. (Danvers, MA, USA), and antibody against pSmad3 from Abcam.

### 5.8. Whole-Cell Extract Preparation

Procedure described in detail in the Appendix A section.

### 5.9. Western Blot Analysis

We used the protocol described in a previous study [48]. All the raw blots have been included in manuscript as Appendix A.

### 5.10. RT-PCR

We used the RNAeasy kit (QIAGEN) and IScript RT-PCR kit (BioRad, Hercules, California, USA) to extract the total RNA and prepare the cDNA, respectively. The qPCR was performed using SYBR^®^ Premix Ex Taq II (Applied biosystem and Life technology, Waltham, MA, USA) and set up in a 7500 Fast Real-Time PCR system (Applied Biosystems, Foster City, CA, USA). The primers were purchased from Integrated DNA technologies (IDT, Coralville, IA, USA).

SMAD7

Forward primer: 5′-CTTCTCCTCCCAGTATGCCA-3′

Reverse primer: 5′GAACGAATTATCTGGCCCCT-3′

Serpine 7 (PAI-1)

Forward primer:5′- CAGCATGTTCATTGCTGCCC-3′

Reverse primer: 5′GGAGAGGCTCTTGGTCTGAAA-3′

PLEXIND1

Forward primer: 5′-GCTGGCCCATTCAAGATCC-3′

Reverse primer: 5′-GCACCAAATGGAAATACTTCTCTGT-3′

### 5.11. Design and Synthesis of Peptides

Described in the Appendix A section.

### 5.12. In Vivo Tumor Models

For all animal studies, we used 6–8-week-old female SCID mice that were purchased from the National Cancer Institute Animal Production Program (MD, USA). Tumor volumes were calculated using the formula *V* = 0.5 × a × b2, where ‘a’ is the longest tumor axis, and ‘b’ is the shortest tumor axis.

### 5.13. Tumor Growth Studies

Protocols for all the in vivo studies are described in the Appendix A section.

### 5.14. Immunohistochemical Staining

Procedure described in detail in the Appendix A section.

### 5.15. Structural Modeling

Procedure described in detail in the Appendix A section.

### 5.16. Statistical Analysis

We performed the log2 transformation of all tumor volume and weight data to form a normally distributed data and used unpaired *t*-tests. Statistical significance was defined as *p* < 0.05, and a high level of statistical significance was defined as *p* < 0.01.

## Figures and Tables

**Figure 1 cancers-13-04048-f001:**
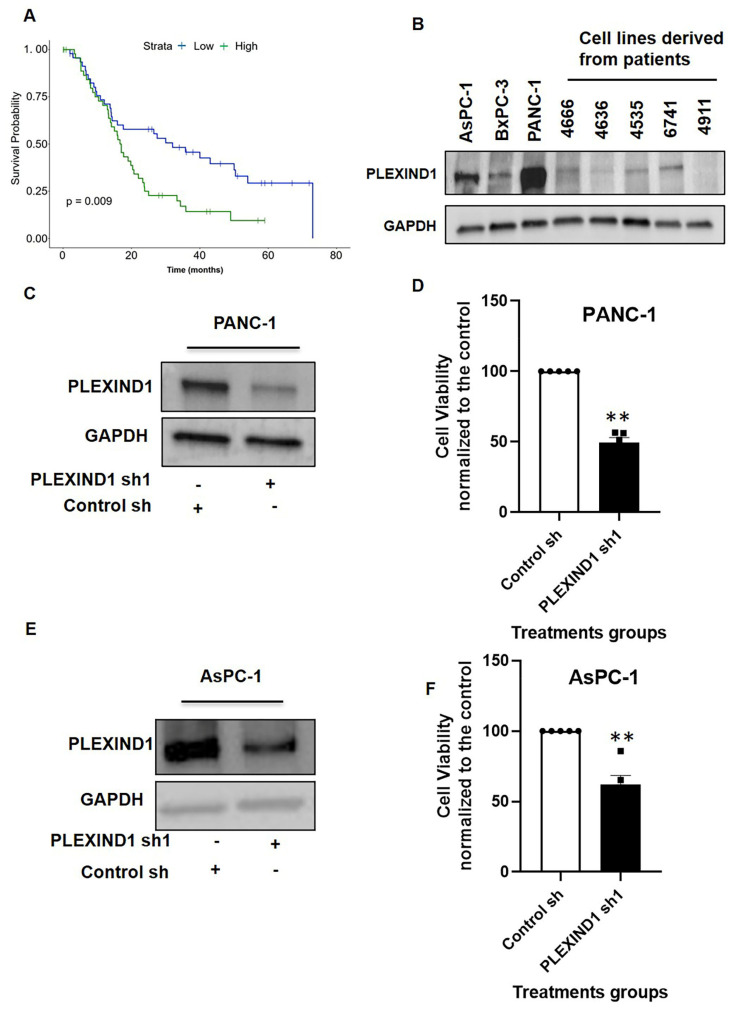
Expression pattern and functional role of PLEXIND1 in PDAC. (**A**): Survival probability analysis of pancreatic ductal adenocarcinoma (PDAC) patients with high and low levels of PLEXIND1 expression. (**B**): Western blot analysis of PLEXIND1 expression in commercial and patient-derived PDAC cell lines. (**C**,**E**): Western blot analysis of PLEXIND1 expression in PDAC cell line PANC-1 and AsPC-1 after shRNA treatment, respectively. (**D**,**F**): Cell viability assay for PANC-1 and AsPC-1 cells with and without PLEXIND1 knockdown grown in 2D cell culture for 72 h. Data are plotted as percentage of control cells. Statistical significance: ** *p* < 0.01 vs. control. Error bars represent standard error of the mean.

**Figure 2 cancers-13-04048-f002:**
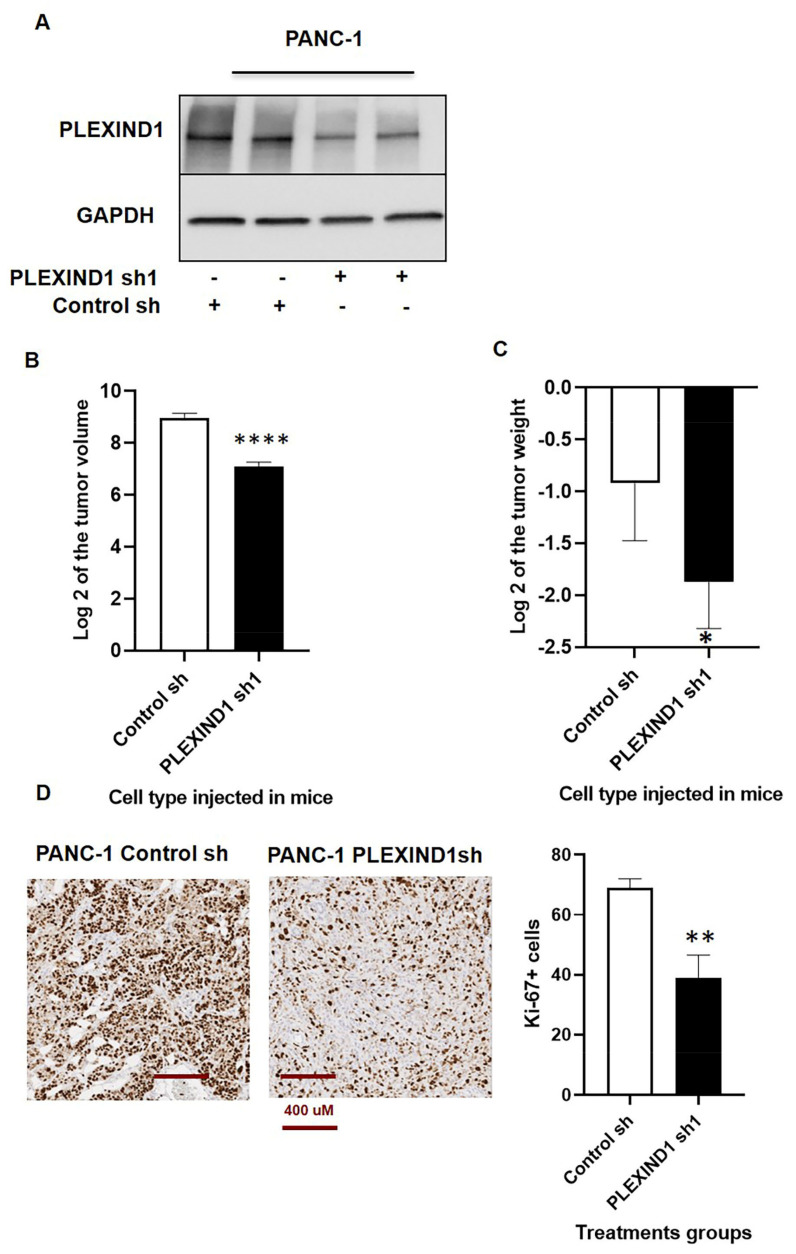
Role of PLEXIND1 in PDAC progression. (**A**): Western blot analysis of PLEXIND1 expression in tissues derived from mice implanted with PANC-1 cells treated with control and PLEXIND1 shRNA 1. (**B**,**C**): Log2 values of tumor volumes and weights from orthotropic mice models implanted with PANC-1 cells with PLEXIND1 knockdown. (**D**): Representative images and quantification of immunohistochemical analysis of Ki-67 in tissues derived from the above mice. Statistical significance * *p* < 0.05 vs control, ** *p* < 0.01 vs control, *** *p* < 0.001 vs control, **** *p* < 0.0001 vs control 2.3. PLEXIND1 and TGFβRII Can Form an Immunocomplex in PDAC.

**Figure 3 cancers-13-04048-f003:**
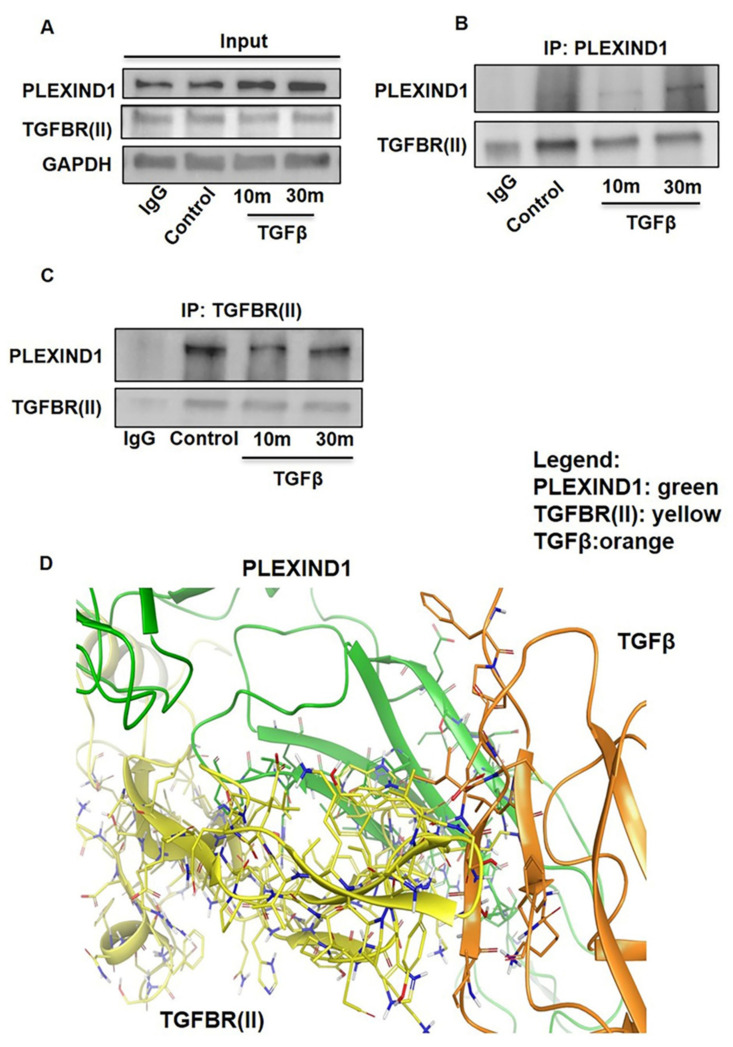
PLEXIND1 and TGFβRII can form an immuno-complex in PDAC. (**A**): Western blot analysis showing the input samples. (**B**,**C**): Co-immunoprecipitation assays showing that PLEXIND1 and TGFBRII are present in the same immunocomplex in PANC-1 cells. (**D**): Computational modeling analysis showing that PLEXIND1 and TGFBRII can form a complex in the presence of TGFβ.

**Figure 4 cancers-13-04048-f004:**
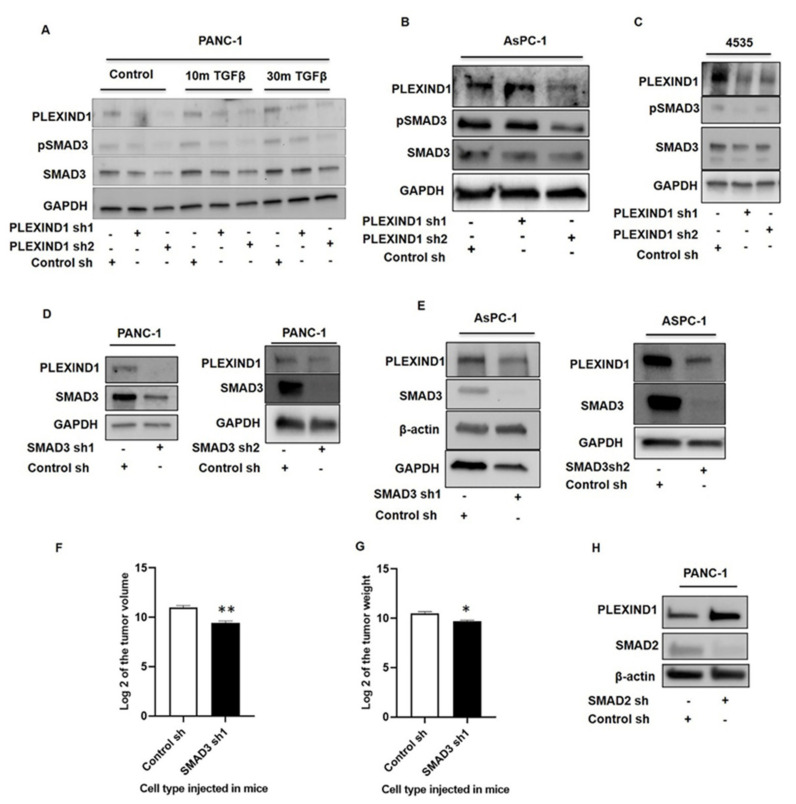
PLEXIND1 modulates SMAD3 signaling and, eventually, PDAC growth. (**A**): Western blot analysis showing levels of phosphorylated and total SMAD3 in PANC-1 cells with reduced PLEXIND1 expression at basal levels and upon TGFβ induction. (**B**,**C**): Western blot analysis showing levels of phosphorylated and total SMAD3 in AsPC-1 and 4535 cell lines. (**D**,**E**): Western blot analysis showing PLEXIND1 expression in PANC-1 andAsPC-1 cells with reduced SMAD3 levels. (**F**,**G**): Log2 values of the tumor volume and weight from orthotropic mice models implanted with PANC-1 cells with SMAD3 knockdown. (**H**): Western blot analysis showing PLEXIND1 expression in PANC-1 cells with SMAD2 knockdown. Error bars represent standard error of the mean. Statistical significance * *p* < 0.05 vs. control, ** *p* < 0.01 vs. control.

**Figure 5 cancers-13-04048-f005:**
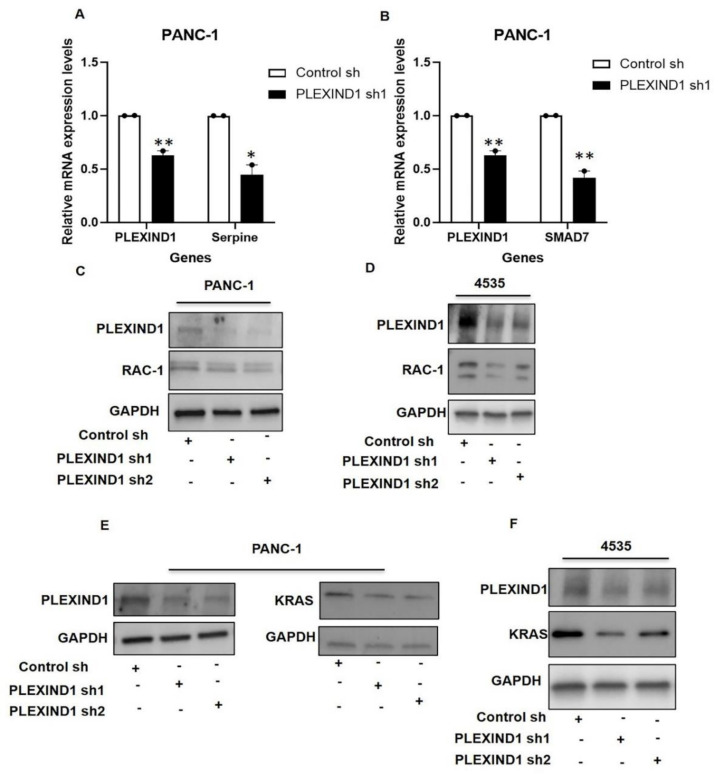
Decreased PLEXIND1 expression results in reduced RAC1 expression in PANC-1 cells. (**A**,**B**): Real-time quantifications of Serpine and SMAD7 expression in PANC-1 cell with PLEXIND1 knockdown. (**C**,**D**): Western blot analysis showing expression of RAC-1 in PANC-1 and 4535 cells with reduced PLEXIND1 levels. (**E**,**F**): Western blot evaluation of KRAS expression in PANC-1 and 4535 cells with PLEXIND1 knockdown. Statistical significance * *p* < 0.05 vs. control, ** *p* < 0.01 vs. control. Error bars represent standard error of the mean.

**Figure 6 cancers-13-04048-f006:**
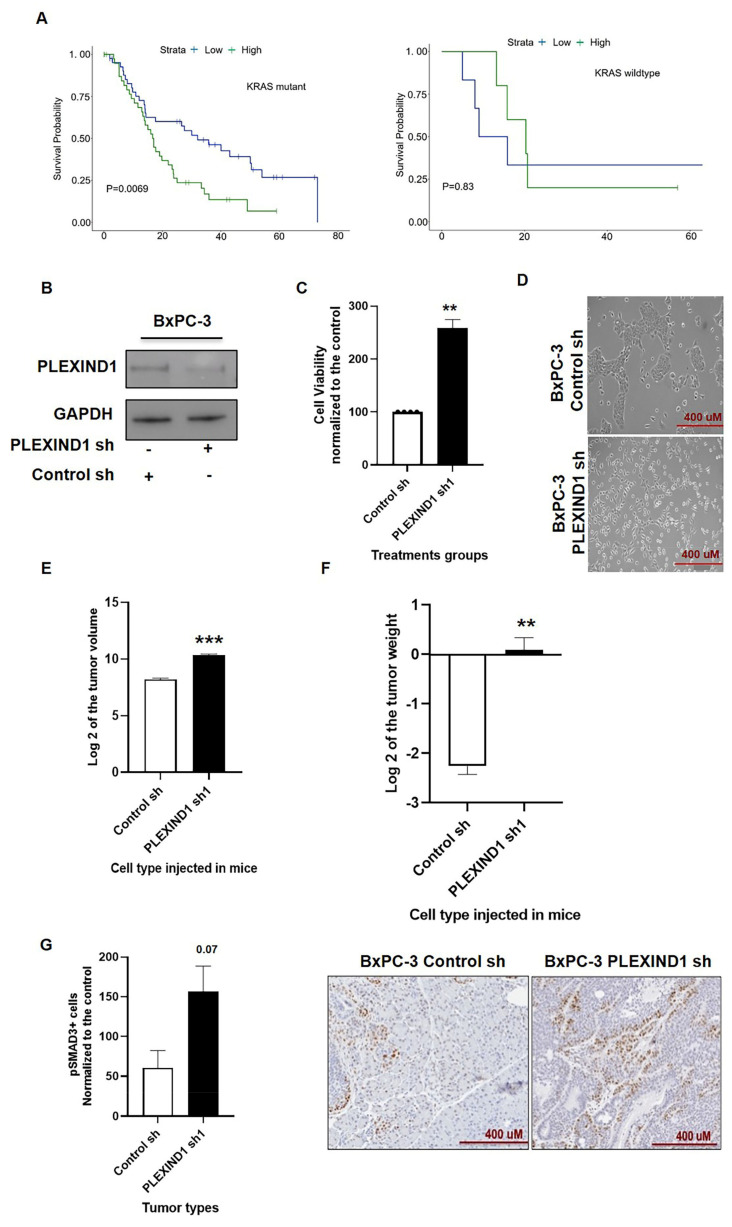
PLEXIND1 acts as a tumor suppressor in KRASwt PDAC cell line BxPC-3. (**A**): Survival probability analysis of pancreatic ductal adenocarcinoma (PDAC) patients with mutant and wild-type KRAS with respect to PLEXIND1 expression. (**B**): Western blot analysis showing reduced expression of PLEXIND1 upon shRNA treatment. (**C**): Cell viability assay for BxPC-3 cells with reduced PLEXIND1 expression grown for 72 h. Data are plotted as a percentage of control cells (transfected with control shRNA). (**D**): Representative images showing the morphological changes in BxPC-3 cells on PLEXIND1 knockdown. (**E**,**F**): Log2 values of tumor volumes and weights from orthotropic mice models implanted with BxPC-3 cells with PLEXIND1 knockdown (**G**): Quantification of digital images for pSMAD3 at 20X magnification (*n* = 3). Significant differences: * *p* < 0.05 vs control, ** *p* < 0.01 vs control, *** *p* < 0.001 vs control, **** *p* < 0.0001 vs control. Error bars represent standard error of the mean.

**Figure 7 cancers-13-04048-f007:**
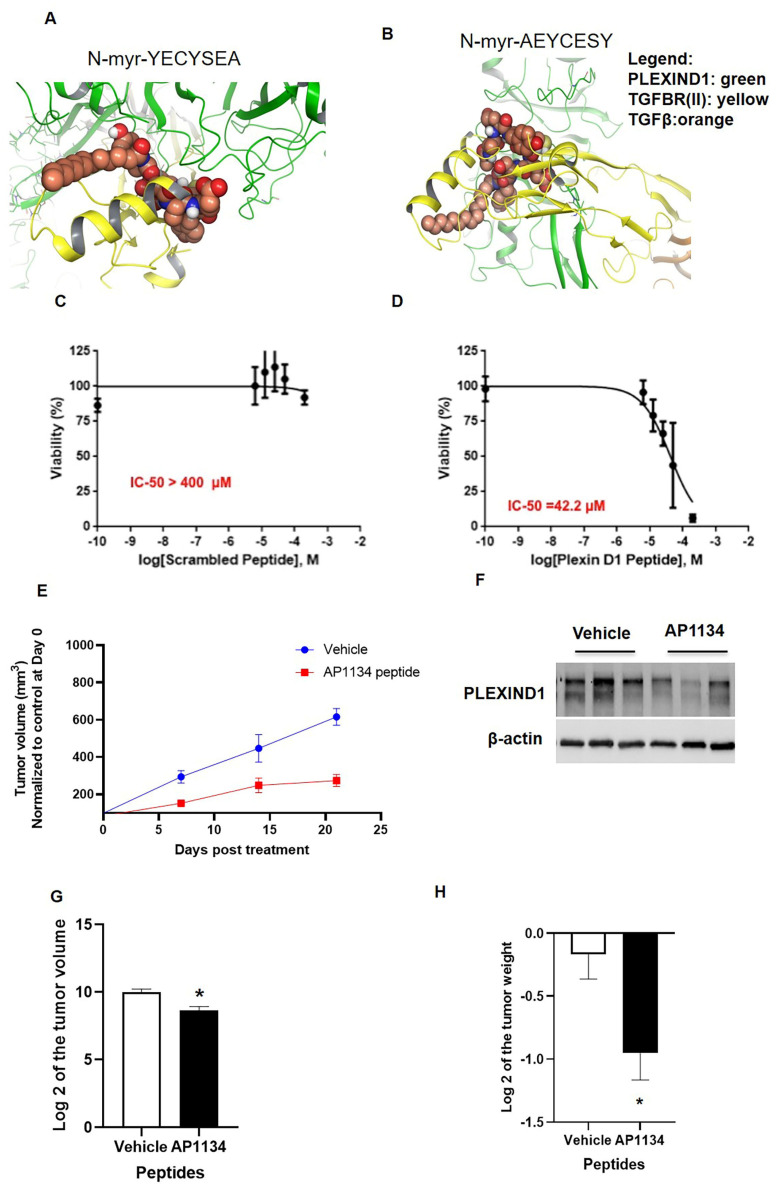
Therapeutic potential of peptide-mediated targeting of PLEXIND1. (**A**,**B**): Computational modeling showing YECYSEA and AEYCESY sequences interacting with PLEXIND1- TFGBRII-TGFβ complex. The peptide is shown in orange carbon “sticks”, and the nearby PLEXIND1 residues are shown in gray carbons (but residues of type Ala or Asp are shown in purple carbon). (**C**,**D**): Cell viability assays for PANC-1 cells treated with either scramble or PLEXIND1 targeting peptide. (**E**): Tumor volume from NSG mice models in which first tumors were grown until about 250 mm3 and then were treated with either vehicle (control) or AP1134. (**F**): Western blot analysis confirming the reduced expression of PLEXIND1 upon AP1134 peptide treatment. (**G**,**H**): Log 2 values of tumor volumes and weights from NSG mice treated with either vehicle (control) or AP1134 at the termination of the experiment. * *p* < 0.05 vs. control. Error bars represent standard error of the mean.

## Data Availability

Not applicable.

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
