# Peer review of "Role of PLEXIND1/TGFβ Signaling Axis in Pancreatic Ductal Adenocarcinoma Progression Correlates with the Mutational Status of KRAS"

_cancers, 2021, doi:10.3390/cancers13164048_

Round 1

Reviewer 1 Report

This is an interesting study clarifying the role of PLEXIND1/TGFβ signaling axis in PDAC. Overall, the study is well-designed and well-performed.

  1. Discussion is included in the Results section (The 1st and 2nd paragraphs of page 12, for example).
  2. Conclusion is too long.
  3. It is unclear why a specific PDAC cell line was used for respective assays.
  4. Description of statistical analysis is insufficient. The authors should describe the analysis for the survival analysis (Fig. 1A, etc).
  5. Number of samples for each assay should be clearly stated.

Author Response

Thank you.

Reviewer 2 Report

The authors present a rather convoluted story that starts with PLEXIND1 and winds through TGFbeta, SMAD2, RAC-1, and finally KRAS.  The interesting and perhaps important finding is that biological activity of PLEXIND1 is different in PANC-1 vs BxPC-3 cells, acting in one case as a tumor growth stimulator and the other as a repressor.  The authors attribute this difference as being due to the KRAS status in these two cell lines – KRASmut vs KRASwt.  But these are two different human pancreatic cancer cell lines, with many genetic differences.  For example, the following differences between these two cell lines was reported in Deer et al, Pancreas 39:425, 2010 PMC2860631.

KRAS

TP53

CDKN2A/p16

SMAD4/DPC4

BxPC-3

WT

220 Cys

WT or HD

HD

PANC-1

12 Asp

273 His

HD

WT

HD=homozygous deletion

It is particularly notable that SMAD4 is reported as a homozygous deletion in BxPC-3, which may or may not be any more related to the differential effects of PLEXIND1 seen in the two cell lines as the KRAS genotype. 

Overall, the experimental design and data presented is inadequate to support a major conclusion of the manuscript related to the influence of KRAS status on PLEXIND1 activity.  Much more rigorous testing of this hypothesis is required, for example with paired isogenic cell lines.

Additional specific comments:

Line 33:  Reference 1 does not relate to the first sentence in the introduction.

Lines 73, 109.  Is it overstating to say that PLEXIND1 is essential for cell viability or tumor growth.  The effects of knockdown are modest.

Figure 1A and 6A.  There is no indication of how the expression data or survival data were obtained, the characteristics of the patients, or other critical information needed to interpret these figures.

There is a lack of a demonstration of reproducibility and a quantitation of effect on the interpretation of western blot data.  For example, Figure 4 D, E, Figure 5 C -F.

Author Response

Thank you.

Round 2

Reviewer 1 Report

Overall, the authors have addressed my concerns.

Author Response

We thank the Reviewer for their time and feedback.

Reviewer 2 Report

The authors made several important improvements to the manuscript but the major point remains:  they attribute the differences in response to KRAS mutational status based on correlative data of KRAS status in cell lines without direct testing of the effects of manipulating KRAS activity.  The results are consistent with an effect of KRAS mutational status but are not proof.  Their response to the previous critique in pointing out the lack of correlation with SMAD4 provides the suggestion that the SMAB4 pathway is not responsible for the observed differences but does not provide additional support for the conclusion in the title of the manuscript that KRAS status is responsible.

In the absence of additional data with KRAS mutational status specifically altered, alterations could be made to the text to insure this level of certainty is clear.  For example:

  • the title could be altered to remove the reference to KRAS influence
  • the simple summary could be altered to suggest that PLEXIND1 functions differently in pancreatic cancer cell lines and the difference correlates with KRAS mutational status (line 22)
  • the abstract could be modified so the last phrase indicates differential roles of PLEXIND1 in cell lines that correlate with KRAS wt vs mut status (line 39)
  • the Introduction altered to remove or soften the conclusion that the genetic status of KRAS could modulate the role of PLEXIND1 (line 68)
  • the conclusion that PLEXIND1 is pro-tumorigenic in KRASmut cells and a tumor suppressor in KRASwt cells in the Discussion needs qualified by saying the results are suggestive of this but are based on a correlation and are not definitive without the direct manipulation of KRAS status in the same cell type. (line 390)
  • The Conclusions must be re-written to accommodate the fact that the data are suggestive but not conclusive with respect to KRAS mutational status. (starting at line 438)
